# LONG-CONTEXT EXTRAPOLATION VIA PERIODIC EXTENSION

## ABSTRACT

Long-context extrapolation aims to extend the contextual window of large language models to process more contextual information, which is widely adopted in industrial applications. Current mainstream solutions involve increasing the rotation base of RoPE Su et al. (2024) to varying degrees or introducing optimization strategies such as "low-frequency extrapolation and high-frequency interpolation", in order to enhance the model's extrapolation capabilities for long context. Actually, these methods alter the representation distribution of positional information by adjusting the rotation frequency of positional encoding, resulting in inevitably disrupt the attention distribution within the original training length range. In this paper, we analyze this phenomenon from a theoretical perspective and propose a long-context extrapolation strategy that preserves the known distribution via periodic extension of high-dimensional positional encoding. Based on this strategy, we design two methods, namely Extra-PE and Extra-MPE, to significantly enhance the models' long-context extrapolation capabilities without disrupting the positional encoding distribution within the original training length. Through extensive experimental results, it is found that the long-context extrapolation method based on periodic extension can enhance the model's capability in extrapolating long-contexts. Specifically, a model fine-tuned on 32k tokens can extrapolate beyond 80k tokens, surpassing the performance of the NTK-32k model and approaching that of the YaRN-64k model. Furthermore, this method demonstrates significantly superior performance in extrapolating extremely long-contexts compared to other methods. Notably, a model fine-tuned on 8k tokens still does not exhibit perplexity explosion when extrapolating to 80k tokens. Additionally, during the fine-tuning process, our approach achieves optimal performance using only one-fourth of the fine-tuning steps (100 steps) compared to the YaRN Peng et al. (2023) method. Secondly, in our comparative experiments, we found that the period in which the model learns a sufficient number of positional encoding has a significant impact on long-context extrapolation capability. Finally, through attention analysis, we discovered that our method can still maintain a stable level of attention at ultra-long distances, with the mean attention value exceeding 0 at these distances.

## 1 INTRODUCTION

In recent years, large language models (LLMs) based on the Transformer Vaswani (2017) has rapidly developed, demonstrating a powerful reasoning ability that has played a very significant role in the fields of natural language processing (NLP). Long-context extrapolation is a fundamental research area in the field of LLMs, referring to the capability of processing and generating text based on an extended context length.

In order to expand the LLMs' context window, it is necessary to increase the positional encoding to enhance the model's ability to understand longer contexts. Currently, Rotary Position Embedding (RoPE) Su et al. (2024) has become the mainstream positional encoding method for large language models, *e.g.*, LLaMA Touvron et al. (2023). However, when the context length exceeds the training text length, directly extrapolating with RoPE does not achieve the expected results Press et al. (2021). With the deepening of research, some research efforts have been proposed for Long-context extrapolation based on RoPE, such as positional interpolation Chen et al. (2023), NTK-

Aware Scaled RoPE Peng & Quesnelle (2023), and the improved methods based on NTK-Aware like Dynamic NTK Emozilla (2023), NTK-by-parts Bloc97 (2023), and YaRN Peng et al. (2023). NTK-series methods are widely adopted in long-context extension ( Young et al. (2024), Touvron et al. (2023), Liu et al. (2024)).These methods seek to expand the positional encoding window by increasing the $base$ value. Another method Liu et al. (2023) involves continuously reducing the $base$ value, which significantly shortens the rotation period of the positional encoding. This allows the positional encoding in high-dimensional space to cover the entire rotation period within a limited training length as much as possible, and this approach performs well in terms of Perplexity (PPL). However, Men et al. (2024) indicates that drastically lowering the base value also has some negative impacts on the model's attention mechanism, specifically by disrupting the long-context decay property of the attention mechanism. By observing how attention scores vary with relative distance within both high-frequency and low-frequency spaces, this paper finds that the low-frequency space of positional encoding is of crucial importance for capturing long-distance attenuation. The specific process is detailed in Appendix A.1. In addition, some work also focuses on achieving Long-context extrapolation by adjusting the attention mechanism Chiang & Cholak (2022); Su (2023).

Previous work has demonstrated good performance in achieving long-context extrapolation, but the current mainstream methods mainly adjust the rotation base (base) or m of ROPE. However, the drawback of this approach is that it alters the distribution of positional encoding within the original training length range of the model, which in turn has a certain impact on the attention distribution within that range. In this paper, firstly, based on the aforementioned issues, we attempt to enhance the long-context extrapolation capability without disrupting the positional encoding distribution within the original training length range of the model. Therefore, we propose a new theoretical perspective for analyzing long-context extrapolation capabilities, and from this perspective, we offer a novel viewpoint on the out-of-distribution (OOD) problem and the classification of current mainstream long-context extrapolation methods. Secondly, from our new theoretical perspective, we propose two long-context extrapolation methods based on periodic extension: Extra-PE (Long-context Extrapolation based on Periodic Extension) and Extra-MPE (Long-context Extrapolation based on Mirrored Periodic Extension). These are currently the first effective out-of-domain long-context extrapolation methods. The core idea of this method is "direct extrapolation for high frequencies and periodic extension for low frequencies," which differs from the "high-frequency extrapolation and low-frequency interpolation" approach proposed by "NTK-aware." Finally, we conducted extensive experiments to verify the effectiveness of the long-context extrapolation methods based on periodic extension. In summary, the main contributions of this paper are as follows:

1. This paper proposes a new theoretical perspective for analyzing long-context extrapolation capabilities.

2. Based on this new theoretical perspective, this paper introduces two long-context extrapolation methods using periodic extension. These methods not only achieve effective extrapolation capabilities but also offer higher training efficiency. Specifically, they are capable of sufficient extrapolation to over twice the original length after only 50 fine-tuning steps. When fine-tuned for 100 steps, the model fine-tuned on 16k tokens achieves optimal extrapolation performance, demonstrating a four-fold improvement in fine-tuning efficiency compared to the current best long-context extrapolation method, YaRN Peng et al. (2023).

3. In terms of extrapolation capabilities, the larger the context window of the fine-tuning data, the more significant the improvement in extrapolation capabilities(Figure 1). The two long-context extrapolation methods based on periodic extension designed in this paper excel at extrapolation on ultra-long-context windows. The model fine-tuned on 32k tokens can effectively extrapolate beyond 80k, with extrapolation performance far surpassing the NTK-32k model and approaching the YaRN-64k model. Moreover, after fine-tuning the model using the Extra-PE method and data of 8k length, the model still does not show significant perplexity explosion at a text length of 80k. Additionally, for models trained with the same specification of fine-tuning data (greater than 16k tokens), the Extra-MPE method demonstrates significantly better extrapolation capabilities on ultra-long-context windows compared to the Extra-PE method, but the Extra-PE method performs slightly better on short-distance text windows.

4. This paper finds that the OOD problem is not the only important factor affecting a model's extrapolation capabilities; simultaneously, the model's learning of the periodic distribution of positional encoding is also crucial.

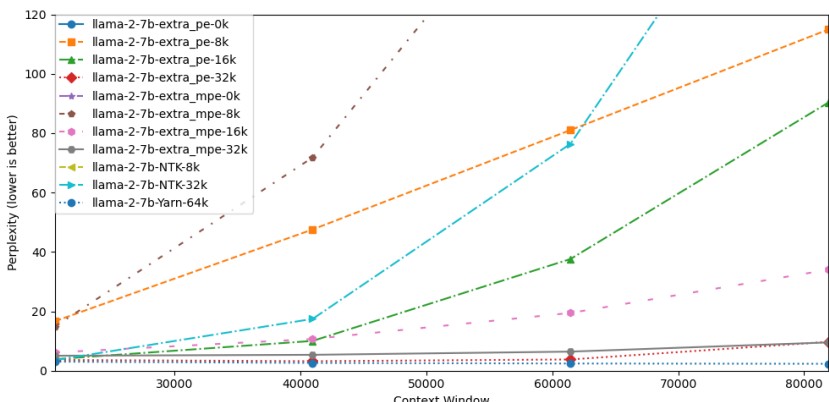

Figure 1: The fine-tuned model with different extrapolation methods exhibits varying PPL (Perplexity) change curves on long-distance context windows.

5. In Appendix A.1 of this paper, various analyses are conducted on the trend of attention scores for the same token as the relative distance increases. It is found that the high-frequency information of positional encoding exhibits significant attenuation at short distances, while the low-frequency information exhibits significant attenuation at long distances. This demonstrates that the long-distance attenuation of attention distribution in ROPE is primarily due to low-frequency positional encoding.

6. The long-context extrapolation method based on periodic extension designed in this paper exhibits significantly different attention score performance compared to other methods (Figure 2). Firstly, it maintains consistency with the original model within the context window of the pre-trained model, enabling the model to prioritize attention on closer similar tokens in short-distance perception. Secondly, this method retains a certain level of attention at ultra-long distances; when the average attention score of other methods at ultra-long distances is 0, the average attention score of this method at ultra-long distances is greater than 0, thus enabling the model to have better ultra-long-distance extrapolation capabilities.

## 2 BACKGROUND AND RELATED WORK

### 2.1 PRELIMINARY

RoPE Su et al. (2024) serves as the theoretical foundation for current long-context extrapolation, which is an improved positional encoding method that has been widely adopted in large language models. RoPE improves upon traditional absolute positional encoding, *e.g.,* sinusoidal positional encoding Vaswani (2017), by aiming to better capture relative position information while avoiding direct modifications to the input vectors. Unlike traditional sinusoidal positional encoding, RoPE does not directly add positional information to the input vectors. Instead, it embeds positional information into the vectors $q$ and $k$, which are the results of transforming the input vector $x$ through the query matrix $Q$ and the key matrix $K$ during the attention mechanism computation. This design preserves the original information of the input vectors, as well as effectively incorporates relative positional information, significantly enhancing the model's performance and robustness.

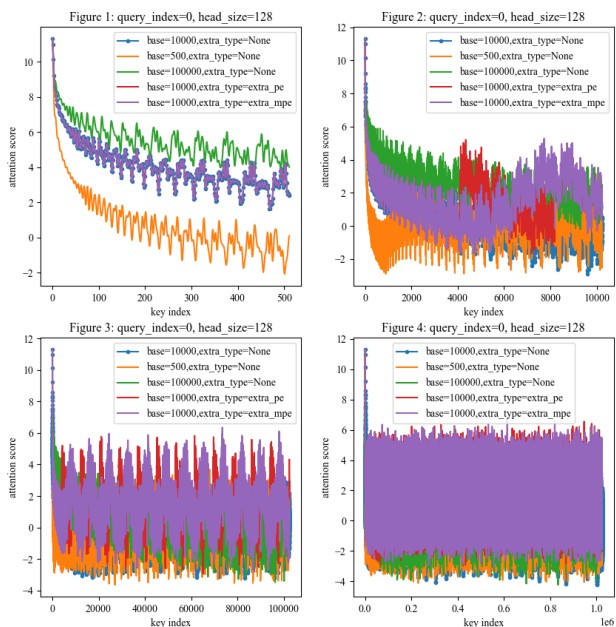

Figure 2: The curves of attention scores for similar tokens, as they increase with relative distance, under different extrapolation methods.

Formally, the transformation process of RoPE in high-dimensional space is represented as:

$$R(m\Theta) =$$
$$\begin{pmatrix}
cos(m\theta_0) & -sin(m\theta_0) & 0 & 0 & \ldots & 0 & 0 \\
sin(m\theta_0) & cos(m\theta_0) & 0 & 0 & \ldots & 0 & 0 \\
0 & 0 & cos(m\theta_1) & -sin(m\theta_1) & \ldots & 0 & 0 \\
0 & 0 & sin(m\theta_1) & cos(m\theta_1) & \ldots & 0 & 0 \\
\vdots & \vdots & \vdots & \vdots & \ddots & 0 & 0 \\
0 & 0 & \ldots & \ldots & \ldots & cos(m\theta_{d/2-1}) & -sin(m\theta_{d/2-1}) \\
0 & 0 & \ldots & \ldots & \ldots & sin(m\theta_{d/2-1}) & cos(m\theta_{d/2-1})
\end{pmatrix}$$
$$(1)$$

where $R(m\Theta)q_m$ is the process of encoding positional information, $\Theta = (\theta_0, \theta_1, \ldots, \theta_{d/2-1}))$ represents the rotation frequencies in each two-dimensional subspace. The magnitude of these rotation frequencies is related to the dimension $i$ as follows:

$$\theta_i = \frac{1}{base^{\frac{2i}{d}}} \quad (2)$$

where $d$ is the total dimension of the positional encoding vector.

RoPE divides the high-dimensional positional encoding vector into $\frac{d}{2}$ subspaces, each consisting of a pair of dimensions. The process of relative positional encoding involves rotating the high-dimensional vectors $q$ and $k$ within these subspaces. The rotation frequency $\theta_i$ for each subspace decreases as the dimension $i$ increases ($base > 1$). The $base$ in Equation 2, which initially is 10000, is treated as a hyperparameter to improve long-context extrapolation. This value is referred to as the rotation base in this paper. Additionally, it is evident from Equation 2 that the high-dimensional space of positional encoding corresponds to a low-frequency space. Conversely, the low-dimensional space is equivalent to a high-frequency space. In this paper, we use the terms high-dimensional and low-dimensional for relevant descriptions.

After the RoPE transformation, the process of obtaining the attention scores for vectors $q$ and $k$ can be expressed as:

$$\text{score}_{m,n} = (R(m\Theta)q_m)^T (R(m\Theta)k_n) = q_m^T R_{n-m}(\Theta) k_n \tag{3}$$

From Equation 3, it can be seen that RoPE achieves the effect of relative positional encoding through absolute positional encoding. The position indices and the base value directly influence the attention scores between the vectors $q$ and $k$. Additionally, these two parameters are also important subjects of current research in long-context extrapolation.

In this paper, we define the maximum context length during the model training process as $L_{\text{train}}$ and the maximum context length during the model inference stage as $L_{\text{inf}}$. The ratio of these two lengths is given by $s = \frac{L_{\text{inf}}}{L_{\text{train}}}$. When $s > 1$, the long-context extrapolation problem arises. To address this task, a straightforward manner, namely PI (Positional Interpolation) Chen et al. (2023), is to implement linear interpolation to compresses the positional index $m$ of RoPE proportionally based on the ratio of the input length during inference to the original maximum context length. By reducing the distance between adjacent tokens, it achieves a certain level of extrapolation. Due to the problem of subpar performance in real-world scenarios of PI, recent researchers have extended the RoPE mechanism to expand the context length. Most of these methods achieve the extension of positional encoding by modifying the $base$ value in Equation 2 and can be categorized into two approaches: increasing the $base$ value and decreasing the $base$ value.

## 2.2 LARGER ROTARY $base$ METHODS

NTK-series method is a set of non-linear interpolation methods. Instead of directly modifying the value of $m$ in Equation 2, these methods achieve the interpolation by increasing the base value in $\theta$. The typical one like NTK-aware Scaled RoPE Peng & Quesnelle (2023), is implemented to let $\theta_i m = \frac{m}{(k\beta)^i}$, and take $\frac{m}{(k\beta)^i} = \frac{m}{s\beta^i}$, which gives $k = s^{\frac{1}{i}}$. Clearly, if $k$ takes the corresponding value in each dimension, it is equivalent to the PI method. This method sets $k$ as a quantity related only to $s$, $i.e.$, let $i = \frac{d}{2} - 1$, then $k = s^{\frac{2}{d-2}}$. The purpose of this is to make the highest dimension of the positional encoding exactly equivalent to the interpolation. However, the disadvantage of this approach is that when we take the second-highest dimension $i = \frac{d}{2} - 2$, there exists: $\frac{n}{(\beta s^{\frac{2}{d-2}})^i} < \frac{n}{s\beta^{\frac{d}{2}-2}}$. This means that with this value of $k$, some dimensions of the positional encoding cannot be fully interpolated.

Therefore, researchers have made several improvements to the NTK method, including NTK-by-part Bloc97 (2023), Dynamic NTK Emozilla (2023), and YaRN Peng et al. (2023). YaRN Peng et al. (2023) also introduces an attention correction factor to mitigate the disruption caused by positional interpolation on attention.

## 2.3 SMALLER ROTARY $base$ METHODS

Increasing the $base$ value is equivalent to interpolation, but decreasing the base value is entirely different. By reducing the $base$, the period of the positional encoding becomes shorter, allowing the model to learn a more diverse distribution of positional encoding during the training phase. Liu et al. (2023) conducted an in-depth study on the process of scaling the $base$ and found that decreasing the $base$ also achieves good extrapolation results. However, in Men et al. (2024), further analysis from the perspective of the attention mechanism was performed on the method of decreasing the $base$. The study found that the process of reducing the base has a certain impact on the attention scores and identified an absolute lower bound for the base value.

Adjusting the $base$ value undoubtedly modifies the original positional encoding distribution. Although this can address the out-of-distribution (OOD) problem for new positional encoding, it can lead to the forgetting of the originally learned positional encoding. Different from the above methods, this paper proposes a method for long-context extrapolation that differs from the aforementioned approaches. It addresses the OOD problem for new positional encoding without altering the original positional encoding distribution.

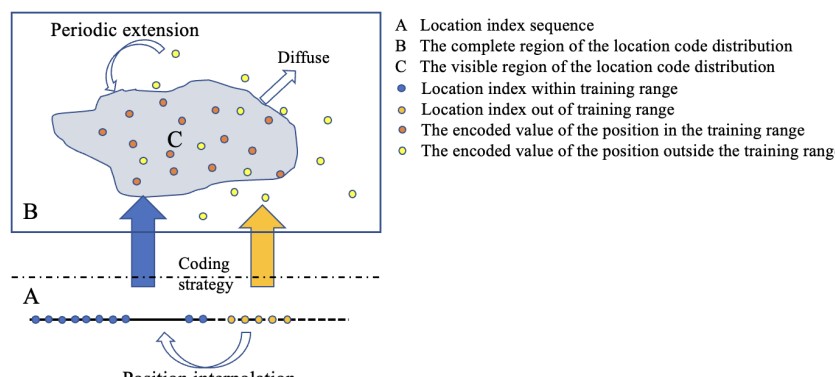

Figure 3: Schematic of positional encoding distribution in a high-dimensional subspace

## 3 THEORY PERSPECTIVE

The OOD problem is widely regarded as a key factor limiting the Long-context extrapolation capabilities of LLMs Ruoss et al. (2023). To address this, many researchers have proposed various positional interpolation methods to confine token position indices within a limited window Chen et al. (2023). This paper presents a new perspective for analyzing Long-context extrapolation capabilities.

Figure 3 represents the distribution of positional encoding in a certain high-dimensional subspace. It is divided into three regions (A, B and C) and four types of data distribution points with different colors. Region A represents the linearly increasing position indices during the encoding process. Regions B and C both show the positional encoding distributions after the ROPE operation, but region B includes the full range of encoding (even beyond the training length), while region C represents the encoding learned within the training length. The areas within B but outside C represent unknown or OOD regions. Differently colored dots illustrate the data distribution across these regions. The reason for describing a high-dimensional subspace is that in a low-dimensional subspace, region C would be infinitely close to region B. In contrast, the high-dimensional subspace is the focus for Long-context extrapolation. When the inference text length exceeds the training text length, the corresponding positional encoding are likely to fall into the unknown areas of region B, leading to the OOD problem.

Below, we use the Figure 3 to categorize and describe the existing mainstream methods for Long-context extrapolation. These methods can be broadly classified into three categories: positional index interpolation within region A, outward expansion from the visible area C of the positional encoding distribution, and inward convergence towards the complete area B of the positional encoding distribution. With this theoretical perspective, direct extrapolation involves making no adjustments for positions that exceed the training text length, relying solely on the good extrapolation properties of ROPE. The result is that a large number of positional encoding fall within the unknown areas of region B in the high-dimensional subspace, leading to very poor extrapolation performance. To address this, there are three main approaches to mitigate this issue as follows.

**Position Interpolation.** PI Chen et al. (2023) involves compressing and adjusting the positions that exceed the training text length within region A. This is done by reducing the spacing between position indices within the training length range and directly inserting the excess position indices into the training length, thereby keeping the positional encoding distribution within the visible area C. Further, researchers have proposed various strategies to adjust the $base$ of RoPE, including NTK Peng & Quesnelle (2023), NTK-by-part Bloc97 (2023), dynamic NTK , and Yarn Peng et al. (2023). These methods are essentially improved versions of PI, using nonlinear interpolation based on different dimensions of the positional encoding.

**Outward Expansion of the Visible Area of positional encoding Distribution.** Randomized positional encoding Ruoss et al. (2023) achieves the expansion of area C by performing random sampling over longer sequences within region A, thereby reducing the likelihood of positional encoding

falling into unknown regions. Similarly, the method of reducing the $base$ involves compressing the rotation periods across all dimensions, allowing the model to learn the positional encoding distribution more thoroughly during the training phase. This effectively expands the visible area of the positional encoding distribution.

**Inward Convergence of the Complete Area of positional encoding Distribution.** The advantage of this approach is that it does not disrupt the visible area C of the model's positional encoding distribution, ensuring that the attention distribution within the training length remains intact. To the best of our knowledge, no such methods currently exist. Analysis of Figure 3 reveals that this approach aims to converge the unknown regions in Area B into the visible region C. To achieve this, we truncate the unknown distributions that have not appeared within the training length in a high-dimensional space, retaining only the unknown code distributions that have been learned within the training length. Based on this, we propose a periodic extension method for long-context extrapolation.

## 4 METHODS

When the dimension of the positional encoding $d > d_{extra}$, it is considered that the positional encoding distribution within that dimensional subspace is incomplete, meaning there are unknown regions as shown in Figure 3.

Motivated by the following two points, this paper designs two long-context extrapolation methods based on periodic extension:

1. Analyze the impact of non-persistent positional encoding strategies on the model training process.

2. Through comparative analysis, verify the primary effects of the two methods on long-context extrapolation capability and long-distance attention distribution.

In this case, we propose two methods via periodic extension shown in Figure 4, *i.e.*, Long-context extrapolation based on periodic extension of the positional encoding distribution (Extra-PE for short) and Long-context extrapolation based on mirrored periodic extension of the positional encoding distribution (Extra-MPE for short). By periodically extending the learned positional encoding distribution from the training process to intervals beyond the $L_{train}$, we effectively erase the positional encoding distribution in the unknown regions.

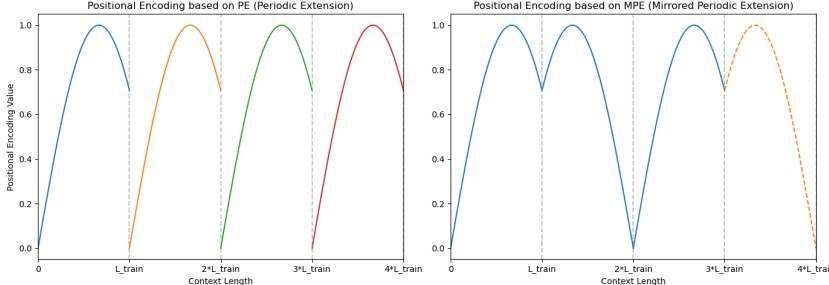

Figure 4: Long-context extrapolation via periodic extension and mirrored periodic extension

### 4.1 EXTRA-PE

Based on the preceding analysis, we only need to apply special handling to dimensions higher than $d_{extra}$, while direct extrapolation can be used for the lower-dimensional subspaces. Therefore, our research focus is on the high-dimensional subspaces greater than $d_{extra}$. We take a high-dimensional subspace $i$ as the subject of our study and define the extrapolation period as $T_{exten}$, the dimension is $dim$, and the length of the input text is $seq_{len}$. The rotational transformation at position $m$ can be represented as $R^{'}(m\theta_i)$:

$$R^{'}(m\theta_i) = \begin{cases} R((m * \theta_i) \bmod T_{exten}), & seq\_len > L_{train} \text{ and } dim \geq \frac{d_{extra}}{2} - 1 \\ R(m\theta_i), & \text{otherwise} \end{cases} \quad (4)$$

where $T_{exten} = L_{train}\theta_i$.

Based on Equation 4, we can achieve the effect shown in the left figure of Figure 4 within the high-dimensional positional encoding space. Although this operation causes periodic overlapping of the positional encoding in the high-dimensional space, it does not perform periodic replication across the entire dimensional space, thus preserving the distinguishability property of the positional encoding.

## 4.2 EXTRA-MPE

Following the above definitions, we continue to derive the process of mirrored periodic extension for the positional encoding distribution in high-dimensional space.

First, we complete the mirroring process of the positional encoding distribution:

$$G(m\theta_i) = \begin{cases} R(m\theta_i), m \in [0, L_{train}) \\ R(T_{exten} - m\theta_i), \ m \in [L_{train}, 2L_{train}) \end{cases} \tag{5}$$

where $T_{exten} = 2L_{train}\theta_i$. Then, the process based on mirrored periodic extension is represented as:

$$H(m\theta_i) = \begin{cases} R(m\theta_i \mod T_{exten}), \lfloor \frac{m}{L_{train}} \rfloor \in \{2n \mid n \in \mathbb{N}\} \\ R((T_{exten} - m\theta_i) \mod T_{exten}), \lfloor \frac{m}{L_{train}} \rfloor \in \{2n + 1 \mid n \in \mathbb{N}\} \end{cases} \tag{6}$$

Equation 6 can be combined as:

$$\begin{aligned} H(m\theta_i) = (\lfloor \frac{m}{L_{train}} \rfloor \mod 2)R((T_{exten} - m\theta_i) \mod T_{exten}) \\ + (1 - \lfloor \frac{m}{L_{train}} \rfloor \mod 2)R(m\theta_i \mod T_{exten}) \end{aligned} \tag{7}$$

Therefore, the final formula for Extra-MPE can be expressed as:

$$R^{'}(m\theta_i) = \begin{cases} H(m\theta_i), seq_{len} > L_{train} \text{ and } dim \geq \frac{d_{extra}}{2} - 1 \\ R(m\theta_i), \text{ otherwise} \end{cases} \tag{8}$$

Through Equations 7 and 8, we can achieve the effect shown in the right sub-figure of Figure 4 within the high-dimensional positional encoding space.

The Extra-MPE method addresses the issue of discontinuity in the positional encoding strategy present in the Extra-PE method, albeit with an increase in computational cost. Additionally, the extrapolation period of Extra-MPE is longer than that of Extra-PE, which means that when using the same length of fine-tuning data, the Extra-PE method can learn more periodic information.

Our method, which involves periodic extension of high-dimensional positional encoding distributions, has the following advantages: it only intervenes when the input text length exceeds $L_{train}$ and the dimension is higher than $d_{extra}$, thus preserving the positional encoding distribution within the training length $L_{train}$ and avoiding the attention field resolution issues caused by the compression of distances between adjacent positions in positional interpolation methods. Additionally, by converging the full positional encoding distribution in high-dimensional subspaces, it addresses the OOD problem. Compared to extending the learning range of high-dimensional positional encoding by reducing the base, this approach theoretically has a higher potential for achieving infinite extrapolation.

## 5 EXPERIMENTS

Based on the above analysis, this paper designs relevant experimental sections. Through experimental testing, it verifies the feasibility of the periodic extension approach in long-context extrapolation capability and its strong extrapolation performance in ultra-long distance windows. Additionally, in various comprehensive ability evaluations, these two methods demonstrate excellent performance, especially in long-context summarization and QA tasks, where they show significant advantages.

## 5.1 Training

For detailed information on model training, please refer to Appendix B, Section 1.1.

## 5.2 Evaluations

In order to better analyze the performance and compare the advantages and disadvantages of the two periodic extension methods mentioned in this paper, specifically in terms of Long-context extrapolation and overall model capability, we need to conduct detailed experimental comparisons and theoretical analyses.

For specific experimental content and analysis results, please refer to Appendix B, Section 1.

## 6 Conclusion

This paper proposes a new perspective for analyzing the extrapolation capability of long-context, and based on this perspective, introduces a novel approach and specific implementation for solving Out-of-Distribution (OOD) problems. We achieve the purpose of long-context extrapolation by periodically extending the positional encoding beyond the length of the pre-training data in high-dimensional space. Additionally, this paper proposes two methods of periodic extension: periodic extension of high-dimensional positional encoding and mirrored periodic extension of high-dimensional positional encoding. Corresponding comparative experiments are conducted for both methods to further verify the effectiveness and advantages of this approach. The specific advantages of this method are summarized as follows:

1.In terms of fine-tuning efficiency, the long-text extrapolation method based on periodic extension designed in this paper is four times as efficient as the YaRN method. That is, we can achieve very good extrapolation results with just a few fine-tuning steps (100 steps).

2.In terms of attention distribution, the long-context extrapolation method designed in this paper does not alter the attention distribution within the pre-training length range of the model, meaning it maintains the model's attention to adjacent tokens at short distances. Additionally, the long-context extrapolation method based on periodic extension demonstrates stronger ultra-long-context extrapolation capabilities. This experimental result corresponds to the conclusion that this method enables the model's attention at ultra-long distances not to decay to zero.

3.Through experimental validation, this paper finds that, besides the OOD problem, the model's ability to learn sufficient periodic positional encoding is also an important factor affecting its long-context extrapolation capability.

4.When comparing the two extrapolation methods designed in this paper, Extra-PE performs better on short-distance context windows and can learn more periodic characteristics with the same length of fine-tuning data. On the other hand, the Extra-MPE method demonstrates stronger ultra-long-context extrapolation capabilities after learning positional encoding information for at least four periodic lengths.

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

# APPENDIX

## A.1 PERSPECTIVE OF ATTENTION DISTRIBUTION

In this paper, by analyzing the changes in attention scores between two identical tokens as the relative distance increases, we theoretically investigate the following two questions:

1. Whether adjusting the base and the Long-context extrapolation method based on periodic extension produce different effects on the attention distribution at short and long distances.

2. Whether the long-distance attenuation observed in the attention distribution when using the ROPE (Rotary Position Embedding) positional encoding method is primarily attributed to the influence of low-frequency spatial positional encoding.

### A.1.1 THE INFLUENCE OF THE BASE ON THE ATTENTION DISTRIBUTION

The changes in attention scores for the same tokens as the relative distance increases, when the base is set to 10000, 500, and 100000, are shown in Figure 6. The experimental results indicate that:

1. Within a short-distance window, adjusting the base will cause significant changes in attention. Increasing the base will enhance the attention between adjacent tokens, while decreasing the base will reduce it.

2. Within an ultra-long-distance window, adjusting the base results in an attention distribution that is consistent with the original base. The attention ultimately oscillates around the mean value of 0.

### A.1.2 THE INFLUENCE OF THE EXTRA-PE AND EXTRA-MPE METHODS ON THE ATTENTION DISTRIBUTION

By comparing the impact of the Extra-PE and Extra-MPE methods on the attention scores as the relative distance increases, as shown in Figure 7, the experimental results indicate that:

1.When using the method based on periodic extension for long-context extrapolation within the pretraining data window of the model, no changes occur in the attention. This performance allows the model to maintain its original ability when processing nearby tokens.

2.When the length of the inference data exceeds the size of the pretraining data window, both the Extra-PE and Extra-MPE methods cause a certain degree of increase in the attention distribution.

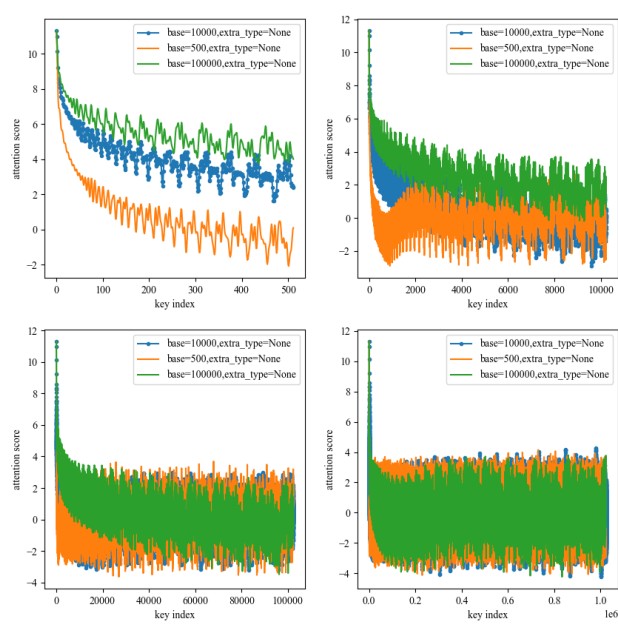

Figure 5: The curve of attention scores between identical tokens as the relative distance increases, when the base takes different values.

Additionally, when the length of the inference data exceeds the length of the pretraining data, the Extra-PE method results in a sudden change in attention.

3.Within ultra-long-distance windows, the method based on periodic extension causes the mean value of the attention to eventually converge to a value greater than 0. This performance enables the model to have better extrapolation performance when extrapolating to longer distances.

### A.1.3 THE IMPACT OF DIFFERENT FREQUENCY SPACES ON THE LONG-DISTANCE ATTENUATION OF ATTENTION

With the base set to 10000, we observe the trend of attention distribution changes in both high-frequency and low-frequency spaces, and further analyze the impact of different frequency spaces on the long-distance attenuation of attention. In this paper, we dissect the 128-dimensional positional encoding and select the first 36 dimensions and the last 36 dimensions of the positional encoding vectors respectively for attention score calculation. At this point, the changes in attention scores for the same tokens as the relative distance increases are shown in Figure 9. The experimental results indicate that:

1. Within a short-distance context window, the positional encoding in the high-frequency space is highly sensitive to the distance between tokens, demonstrating a clear 'short-distance attenuation' effect on attention. whereas, the positional encoding in the low-frequency space is relatively insensitive to distance.

2.In the long-distance context window, the positional encoding in the high-frequency space becomes insensitive to the distance between tokens, and the attention distribution exhibits continuous and non-decaying oscillations. In contrast, the positional encoding in the low-frequency space shows a significant attenuation as the relative distance increases. Therefore, the positional encoding in the low-frequency space is an important factor contributing to the 'long-distance attenuation' of attention within the entire encoding space.

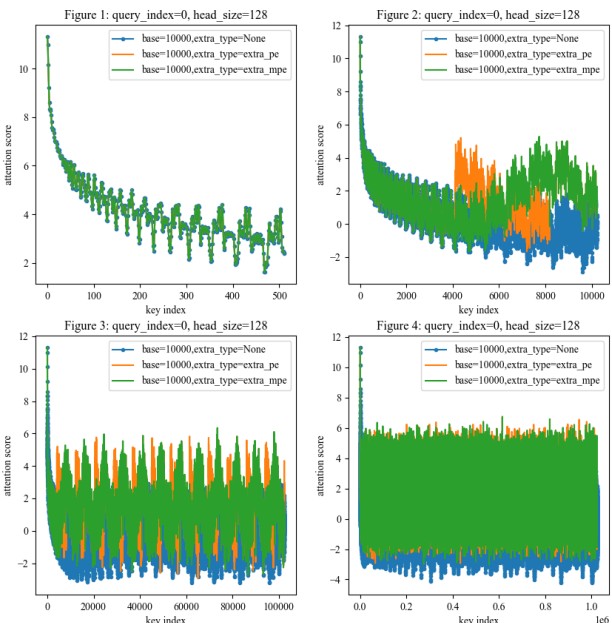

Figure 6: The curve of attention scores between identical tokens as the relative distance increases when implementing Long-context extrapolation using the Extra-PE and Extra-MPE methods.

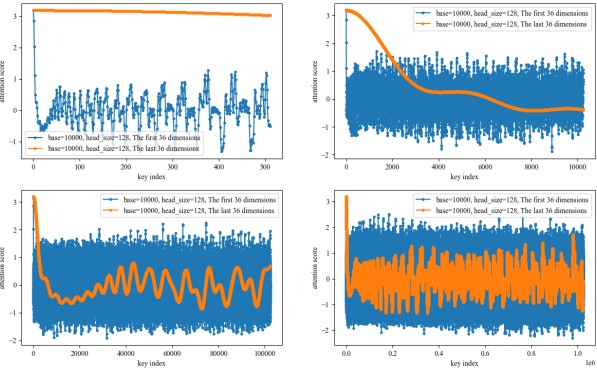

Figure 7: The variation curve of attention scores within high-frequency and low-frequency spatial domains.

3.Within an ultra-long-distance context window, regardless of whether it is the positional encoding in the high-frequency space or the low-frequency space, the attention exhibits oscillating behavior with a mean value of 0 and without continuous attenuation. However, the amplitude of the oscillations in the low-frequency space is smaller than that in the high-frequency space.

## B.1 EXPERIMENTAL CONTENT

### B.1.1 TRAINING

The base model for this method is Llama2-7b Touvron et al. (2023), with a data context window size of 4k during its training phase. The model training process is completed based on the YaRN Peng et al. (2023) training framework. Additionally, in terms of model structure, only the strategy for implementing positional encoding was adjusted accordingly, without modifying other aspects of the model structure or introducing attention adjustment factors in the process of calculating the attention score. To test the impact of the number of model training steps and the context length of the model fine-tuning data on the model's extrapolation performance, the model training was conducted based on two dimensions: firstly, the number of training steps was set to 50, 100, 200, and 400, respectively; secondly, the context length of the model fine-tuning data was set to 8k, 16k, and 32k, respectively. All experiments are conducted on a machine equipped with 8 NVIDIA A100 GPUs, each with 40GB of memory. In terms of hyperparameters, we fixed the learning rate at 2e-5, used the Adam optimizer, set the global batch size for model training to 64. Additionally, we leveraged the deepspeed framework Rasley et al. (2020) to achieve efficient model training, and utilized PyTorch Paszke et al. (2019) with Fully Sharded Data Parallelism Zhao et al. (2023) and Flash Attention 2 Dao (2023), where the zero_optimization adopts ZeRO3 strategies Jackson et al. (2010). In terms of the dataset, we adopted the same training set as the YaRN Peng et al. (2023) method. Specifically, the model was fine-tuned based on the PG19 dataset Rae et al. (2019). During the model training process, we validated Motivation 1 in Section 4 and found that the use of non-contiguous positional encoding did not lead to failure in the model training process.

### B.1.2 PERPLEXITY EVALUATIONS

During the model PPL (Perplexity) evaluation phase, we utilize the Proof-pile Azerbayev et al. (2022) dataset, which comprises numerous long sequence samples. From this dataset, we have selected 10 samples as the test set for perplexity evaluation, with each sample containing at least 100K tokens in length. For the perplexity evaluation, we employ the sliding window method introduced by Press et al. (2021). However, to more accurately assess the model's perplexity on long sequence texts, this paper does not set a specific size for the sliding window or a sliding step. Instead, we adopt a truncation method, where the perplexity calculation is performed in one go within the entire truncated text window, i.e., truncate=True.

Firstly, we tested the model's perplexity performance within a context window of 16k tokens as the window size increases. We truncated the data samples sequentially from 2k tokens to 16k tokens, with a step size of 2k tokens each time.

In order to better analyze the performance and advantages/disadvantages of the two periodic extension methods mentioned in this paper in terms of long-context extrapolation and overall model capability, we need to conduct detailed experimental comparisons and theoretical analyses. The specific evaluation content mainly includes three aspects: Perplexity Evaluations, The long-context Retrieval Capability, and The Long Context Understanding Capability.

Figure 8 presents a comparison of the PPL results for the Llama2-7B model extrapolated using Extra-PE, Extra-MPE, "NTK-aware" Peng & Quesnelle (2023), and YaRN methods Peng et al. (2023), respectively. The results indicate:

1.Without fine-tuning on new data, directly using the Extra-PE or Extra-MPE methods yields relatively poor results. This implies that merely addressing the OOD (Out-of-Distribution) problem does not directly lead to a comprehensive improvement in Long-context extrapolation.

2.During the fine-tuning stage, the model must learn positional encoding for at least 4 cycles (i.e., the context length of the fine-tuning data should be at least 4 times the maximum length of the pre-training data) for there to be a significant improvement in extrapolation capability. This suggests that enabling the model to learn the periodicity of the positional encoding distribution is crucial for significantly enhancing its long-context extrapolation ability.

3.Within short-distance context windows, the perplexity performance of the Extra-PE method is superior to that of the Extra-MPE method.

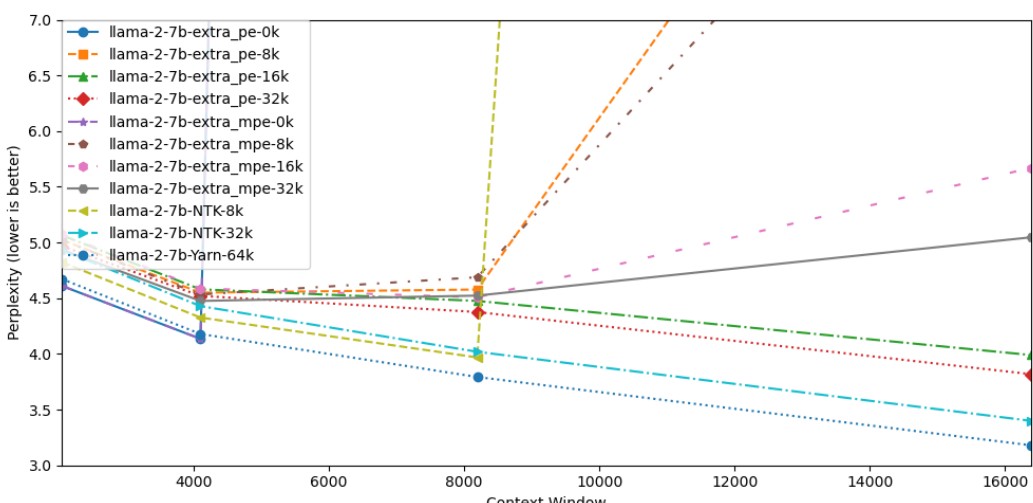

Figure 8: The fine-tuned model with different extrapolation methods exhibits varying PPL (Perplexity) change curves on short-distance context windows.

Secondly, we further tested the model's perplexity performance within a long-context window of 80k tokens as the window size increased. We truncated the data samples in increments of 10k tokens, ranging from 20k tokens to 80k tokens. The results are shown in Figure 1. The experimental results indicate that:

1.Long-context extrapolation methods based on periodic extension excel at ultra-long-context extrapolations.Specifically, the model fine-tuned using the 8k context and the Extra-PE method does not experience a significant perplexity explosion when extrapolated to 80k tokens. Moreover, the model fine-tuned using the 32k context outperforms the NTK-32k Peng & Quesnelle (2023) model by a wide margin and approaches the performance of the YaRN-64k Peng et al. (2023)model.

2.When using the same size of fine-tuning data window, the Extra-MPE method performs better in ultra-long-context extrapolation compared to the Extra-PE method.

Finally, we conducted comparative tests to evaluate the extrapolation capabilities of models based on both the Extra-PE and Extra-MPE methods when different numbers of fine-tuning steps were employed. The results are shown in Figure 6, and we found that:

1.After 50 fine-tuning steps, the model already possesses certain extrapolation capabilities, allowing it to extrapolate to more than twice its original context length.

2.The long-context extrapolation capabilities of the models do not improve with an increase in the number of fine-tuning steps. Both the Extra-PE and Extra-MPE models achieve optimal performance at 100 fine-tuning steps. In terms of fine-tuning efficiency, they significantly outperform the YaRN Peng et al. (2023) method, which requires 400 fine-tuning steps.

When using the Extra-MPE method, the context window of the fine-tuning data must be at least 16k or more; otherwise, the model's extrapolation performance will be relatively poor. This is because within a context window of 8k, the model can only learn a low-frequency positional distribution for one cycle.

### B.1.3 THE LONG-CONTEXT RETRIEVAL CAPABILITY

In terms of the model's long-context retrieval capability, we choose the Long-eval benchmark from Dacheng Li* & Zhang (2023) for evaluation. This paper primarily conducts comparative experiments on various models focusing on the lines task, observing and analyzing the trend of how Accuracy changes as the text length increases.

Table 1 presents the evaluation results of the long-eval benchmark, which indicate:

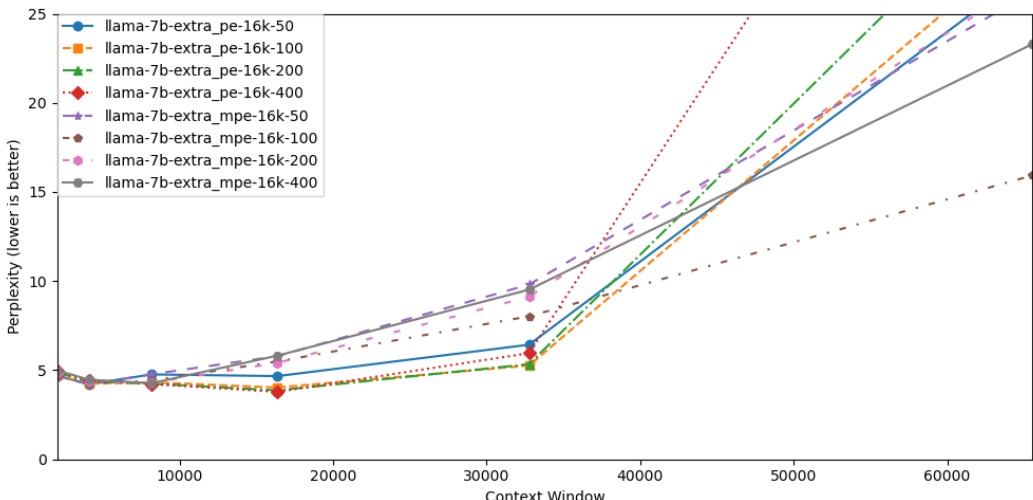

Figure 9: On a context length of 16k, we fine-tune the Llama2-7B model using the Extra-PE and Extra-MPE methods, and plot the PPL (Perplexity) curves based on different numbers of fine-tuning steps.

After fine-tuning with either the Extra-PE or Extra-MPE method, the model exhibits significant improvement in long-context retrieval tasks. Among the models fine-tuned with 8k contexts, the Extra-PE method outperforms the YaRN-8k Peng et al. (2023) model and also surpasses the Extra-MPE-8k model. This demonstrates that the periodic extension method has certain advantages in long-text retrieval tasks, while also indirectly indicating that the model's ability to learn sufficient low-frequency positional encoding periods is equally important for long-text retrieval capabilities.

The experimental results show that the Extra-MPE method performs better in distant-range retrieval compared to near-range retrieval, while the Extra-PE method excels in near-range retrieval. Furthermore, this conclusion reinforces the notion that the Extra-MPE method is more suitable for distant-range tasks compared to the Extra-PE method.

### B.1.4 THE LONG-CONTEXT UNDERSTANDING CAPABILITY

This paper continues to conduct evaluations on tasks related to long-context understanding using a multitask approach based on the Long Bench Bai et al. (2023). The benchmark includes five types of tasks: Single-Doc QA, Multi-Doc QA, Summarization, Few-shot Learning, and Code Completion. The corresponding test datasets for these tasks are, in order: multifieldqa_en Fabbri et al. (2019), 2wikimqa Ho et al. (2020), multi_news Fabbri et al. (2019), Trec Li & Roth (2002), samsum Gliwa et al. (2019).

Table 2 presents the test results of the Long Bench, which indicate that the model fine-tuned using the periodic extension method performs well across the overall tasks, particularly exhibiting significant advantages over the NTK Peng & Quesnelle (2023) and YaRN Peng et al. (2023) methods in the multi-document QA task involving long-contexts and the summarization task.

Table 1: The evaluation results of the long-eval benchmark.

| Model | 5300 | 7600 | 10000 | 12000 | 14500 | 16400 |
|---|---|---|---|---|---|---|
| llama-2-7b | 0 | 0 | 0 | 0 | 0 | 0 |
| llama-2-7b-extra_pe-8k | 0.14 | 0.18 | 0 | 0 | 0 | 0 |
| llama-2-7b-extra_pe-16k | 0.12 | 0.14 | 0.06 | 0 | 0.02 | 0 |
| llama-2-7b-extra_pe-32k | 0.16 | 0.06 | 0.06 | 0.04 | 0.04 | 0.02 |
| llama-2-7b-extra_mpe-8k | 0.08 | 0.08 | 0 | 0 | 0 | 0 |
| llama-2-7b-extra_mpe-16k | 0.14 | 0.08 | 0.16 | 0.02 | 0 | 0 |
| llama-2-7b-extra_mpe-32k | 0.14 | 0.02 | 0.06 | 0.14 | 0.12 | 0 |
| llama-2-7b-NTK-8k Peng & Quesnelle (2023) | 0.06 | 0 | 0.04 | 0.06 | 0.06 | 0.1 |
| llama-2-7b-Yarn-8k Peng et al. (2023) | 0.1 | 0.1 | 0 | 0 | 0 | 0 |
| llama-2-7b-Yarn-64k Peng et al. (2023) | 0.24 | 0.22 | 0.14 | 0.22 | 0.14 | 0.04 |

Table 2: The evaluation results of the long-bench benchmark.

| Model | Avg | multifieldqa_en (Single-Doc QA) | 2wikimqa (Multi-Doc QA) | multi_news (Summ.) | Trec (Few-shot) | samsum (Code) |
|---|---|---|---|---|---|---|
| Yarn-8k Peng et al. (2023) | 31.348 | 29.38 | 9.38 | 9.56 | 64.5 | 43.92 |
| Yarn-64k Peng et al. (2023) | 32.662 | 26.76 | 9.36 | 14.2 | 68.67 | 44.32 |
| Extra-PE-8k | 33.132 | 23.27 | 11.82 | 22.81 | 65 | 42.76 |
| Extra-PE-16k | 32.866 | 27.56 | 12.67 | 18.71 | 64.42 | 40.97 |
| Extra-PE-32k | 31.664 | 24.71 | 10.9 | 15.93 | 66.03 | 40.75 |
| Extra-MPE-8k | 30.692 | 25.59 | 12.24 | 20.95 | 55.94 | 38.74 |
| Extra-MPE-16k | 28.762 | 24.61 | 11.01 | 18.91 | 61.49 | 27.79 |
| Extra-MPE-32k | 27.688 | 23.33 | 9.96 | 12.97 | 56.25 | 56.25 |

