# OpenReview forum: "Long-context Extrapolation via Periodic Extension"
_ICLR.cc/2025/Conference — ICLR 2025 Conference Withdrawn Submission_

### Official Review · Reviewer_u3cx · 2024-11-01

**Soundness:** 1
**Presentation:** 1
**Contribution:** 1
**Rating:** 1
**Confidence:** 2

**Summary:**

This paper provide a theoretical understanding of the essence of long-context extrapolation through RoPE. Two methods, Extra-PE and Extra-MPE, are proposed to enhance the models' long-context extrapolation capabilities.

**Strengths:**

N/A

**Weaknesses:**

This paper is far from ready to be accepted as a conference paper.

1. It is hard to following the author's introduction to the theoretical results.

2. I've never seen a paper that puts all its experimental results in the appendix and say nothing in the main text.

3. The experimental results in Table 1 and 2 show that Extra-PE and Extra-MPE is inferior to the baselines.

4. This paper uses \cite{} on all the citations. Instead, for citations that do not serve as parts of the sentence structure, \citep{} should be used to enclose the citation in parentheses.

**Questions:**

N/A

---

### Official Review · Reviewer_cSmY · 2024-11-02

**Soundness:** 2
**Presentation:** 1
**Contribution:** 2
**Rating:** 3
**Confidence:** 2

**Summary:**

This paper introduces two novel methods to enhance long-context extrapolation capabilities through periodic extension, i.e. Extra-PE and Extra-MPE. Both approaches are backed by theoretical insights and experimental results.

**Strengths:**

P1. The two proposed methods extend the LLM's long-context extrapolation abilities without altering the distribution of positional encoding of the original training length. This attribute, as detailed in Section 4, ensures that the model maintains its ability to process and understand text within the training length while also improving its ability to handle longer contexts. The core implementation of the periodic extension concept in Equation (4) on page 7 is straightforward.

P2. One of the strengths of this paper is the combination of theoretical analysis and experimental validation. The theoretical insights in Section 3 provide an in-depth understanding of long-context extrapolation mechanics, while the experiments (mostly in Appendix) explore multiple facets, including fine-tuning steps, context lengths, and performance metrics such as perplexity.

**Weaknesses:**

C1. The paper's structure could be improved for better readability and comprehension; for example, the section on related work should be condensed to provide more space for the experimental results, which are more crucial to the paper.

C2. Several key baselines are missing. While the paper compares the proposed methods with several baselines, it misses out on including some key variants that could provide a more comprehensive comparison. For instance, in Figure 3, if Yarn-64k demonstrates the best performance, why are Yarn-32k, Yarn-16k, and Yarn-8k not included for a more equitable comparison with the proposed methods?

C3. Some unusual experimental results lack explaination. For example, in Table 2 from Appendix B.1.4, Extra-PE-8k surpasses all longer-context variants by a significant margin, yet no analysis is provided to clarify this observation.

**Questions:**

Why do authors put main experimental results in Appendix, and put Equation 1 in the main body?

---

### Official Review · Reviewer_G5eT · 2024-11-04

**Soundness:** 2
**Presentation:** 1
**Contribution:** 1
**Rating:** 3
**Confidence:** 3

**Summary:**

The paper investigate the long-context extrapolation, where the language models perform inference on data whose context length exceeds maximum length during training. The papers starts with an extensive review of existing extrapolation methods that fall under two categories:
+ Compressing positional encoding of out-of-distribution samples under maximum length during training;
+ Expand the seen-during-training region by reducing the base value of Rotary Positional Embedding.

Motivated the flaw of existing method: distribution of in-distribution will be disrupted during inference, the paper proposed the two encoding methods:
+ Periodic Extension for Long-Context Extrapolation: The paper introduces a novel approach for extending positional encoding beyond the training length by using periodic extension. This approach maintains the original positional encoding distribution within the model's training length and enables effective extrapolation for sequences much longer than the model's original training context.
+ Extra-PE and Extra-MPE: The authors propose two variants of periodic extension: Extra-PE, which periodically extends high-dimensional positional encodings, and Extra-MPE, which introduces a mirrored periodic extension for smoother transitions. Both methods improve long-context extrapolation without disrupting attention within the original training range, with Extra-MPE particularly excelling at ultra-long-context extrapolation.
With benchmarking on real data, the paper demonstrate that the propose PE methods lead to stronger performance and lower perplexity in long-context extrapolation, despite the increased computing cost.

**Strengths:**

+ The paper discusses long-context extrapolation, an important topic that has important application in real world AI systems.

+ The paper proposed two method well motivated on extensive review of existing work.

**Weaknesses:**

+ Non-standard Organization: The paper devotes nearly half of the main text to reviewing and summarizing existing work, while key experimental and evaluation details are relegated to the appendix. Although a thorough discussion of prior research and context can be valuable, it should not overshadow the main contributions and results. The current structure makes it difficult to clearly identify the primary contributions and findings of the paper.

+ Risk of Ambiguity in Positional Encoding with Periodic Extension: The periodic extension method introduces a potential risk of ambiguity in positional encoding that is not addressed in the paper. When out-of-length tokens receive repeated positional encodings, the model may interpret these tokens as if they were at earlier positions in the text. This risk of confusing distant tokens with those at the start of the sequence, particularly in long sentences, is not discussed, nor is any mitigation strategy provided.

+ Unclear Performance Improvement: The performance improvement of the proposed methods (Extra-PE and Extra-MPE) is not clearly demonstrated. In the context-perplexity figures in Appendix B.1.2 and B.1.3, key baseline methods are either missing or show lower perplexity than the proposed methods. The text also primarily compares Extra-PE and Extra-MPE, which are two variants of the same method, without a clear comparison to external benchmarks.

+ Clarity and Coherence in Writing: The paper could improve in clarity and coherence. For instance, terms such as `d_extra` in Section 4 are not defined prior to their use, and Appendix B.1.4 does not explain the evaluation metrics shown in its table, making it challenging for readers to interpret the results.

**Questions:**

+ Please clarify whether the proposed periodic extension assigns the same positional encoding values to different tokens within or beyond  $L_{train}$ . If so, consider adding a theoretical analysis explaining why this does not pose a significant risk, or provide an attention map analysis to demonstrate that repeated positional encoding values do not disrupt the attention pattern by confusing out-of-distribution positions with in-distribution ones.

+ Please adjust Figures 8 and 9, along with the corresponding text, to emphasize the performance advantages of the proposed method. Consider reducing the number of settings included so readers can more easily see the method-wise differences.

+ Consider making Sections 2 and 3 more concise to allocate more space to Section 5, which is central to the main text. Additionally, please correct minor notation and term definition issues, such as those mentioned in the weaknesses section.

---

### Official Review · Reviewer_oYeh · 2024-11-04

**Soundness:** 2
**Presentation:** 1
**Contribution:** 2
**Rating:** 3
**Confidence:** 3

**Summary:**

This paper aims to extrapolate the positional embeddings for Transformer models. Specifically, the authors propose two kinds of extrapolation: ExtraPE and ExtraMPE. They test the method on Llama 2 and show that they are effective to a certain extent.

**Strengths** \
The paper is well-motivated. The research topic is also important for Transformer models.

**Weaknesses**
1. The paper is particularly not well-written.
2. Lacks theoretical justification.
3. The experiments are limited.


In summary, this paper is not well-written and requires significant revision and proper comparisons.

**Strengths:**

The paper is well-motivated. The research topic is also important for Transformer models.

**Weaknesses:**

1. The paper is particularly not well-written. Here are the details (sorted by occurrence rather than importance):
 * Figure 1. (1) A lot of the curves are entangled at the bottom, making them indistinguishable. (2) The colors are repeated (first and last). (3) Numerous curves are informativeless. It is not helpful to show the proposed methods worsening the perplexity by a large margin. (4) The proposed methods have context lengths from 0 to 32K, whereas Yarn (one of the baselines) is fixed to 64K. They are not comparable.
 * Figure 2. All figures (except for the upper left) are not readable because all other curves are under the purple lines.
 * Section 3 “Theory Perspective” has no theory. There is only a hypothesis based on intuition.
 * Two different methods are proposed in Section 4. However, there is no clear motivation for each design. It is also unclear which one should be used in real-world applications because the generation length is not determined beforehand.
 * Section 5: all experiments are in the Appendix while leaving more than 1 page unused in the main text.
2. Lacks theoretical justification. It should be noted that a theory should be a rigorous justification instead of intuition.

3. The experiments are limited. The authors mainly rely on perplexity as the evaluation metric. However, it might not directly correlate with real-world experience. The authors additionally tested on LongBench in the Appendix, but it is not convincing because they selected a subset of LongBench tasks.

**Questions:**

See weaknesses.

---

### Note · Authors · 2024-11-25

I have read and agree with the venue's withdrawal policy on behalf of myself and my co-authors.